# Coronary Stent Thrombosis in COVID-19 Patients: A Systematic Review of Cases Reported Worldwide

**DOI:** 10.3390/v14020260

**Published:** 2022-01-27

**Authors:** Wojciech Jan Skorupski, Marek Grygier, Maciej Lesiak, Marta Kałużna-Oleksy

**Affiliations:** Chair and 1st Department of Cardiology, Lord’s Transfiguration Clinical Hospital, Poznań University of Medical Sciences, 61-848 Poznań, Poland; mgrygier@wp.pl (M.G.); maciej.lesiak@skpp.edu.pl (M.L.); marta.kaluzna@wp.pl (M.K.-O.)

**Keywords:** COVID-19, stent thrombosis, coronary artery, SARS-CoV-2

## Abstract

Approximately 5 million percutaneous coronary interventions are performed worldwide annually. Therefore, stent-related complications pose a serious public health concern. Stent thrombosis, although rare, is usually catastrophic, often associated with extensive myocardial infarction or death. Because little progress has been made in outcomes following stent thrombosis, ongoing research is focusing on further understanding the predictors as well as frequency and timing in various patient subgroups. Coronavirus disease-2019 (COVID-19), a viral illness caused by the severe acute respiratory syndrome-coronavirus-2 (SARS-CoV-2), activates inflammatory mechanisms that potentially create a prothrombotic environment and increases the risk of local micro thromboembolism and all types of stent thrombosis. In-stent thrombosis occurrence increased during the COVID-19 pandemic, however, there is still lack of comprehensive studies describing this population. This review and worldwide analysis of coronary stent thrombosis cases related to COVID-19 summarizes all available data.

## 1. Introduction

Approximately 5 million percutaneous coronary interventions (PCI) are performed worldwide annually. Therefore, stent-related complications pose a serious public health concern [1]. Stent thrombosis is a rare but usually catastrophic event, frequently associated with large myocardial infarction (MI) or death [2,3]. Due to improvements in stent implantation technology and more effective antiplatelet therapy, stent thrombosis continues to occur at a relatively low estimated rate varying between 1–4% [1,4,5,6]. It can be classified as early (within 30 days after stent implantation, late (between 1 month and 1 year after implantation), or very late (>1 year after implantation) (Table 1) [7]. Stent thrombosis can also be classified as definite, probable, or possible on the basis of its clinical presentation [7]. Multiple patient-related, lesion-related, device-related, and procedure-related predictors of stent thrombosis have been described [1,8]. Higher predisposition to thrombosis, both arterial and venous, during COVID-19 has been thoroughly investigated in recent months [9]. Importantly, new studies and registers show that the percentage of stent thrombosis increased during the COVID-19 pandemic, even up to 8.1–21%, however, the study populations were relatively low [10,11,12]. COVID-19, due to direct and indirect effects, creates a significant burden on the cardiovascular system. There is proof that active COVID-19 increases the risk of acute MI [13,14]. Recent reports indicate evidence of myocardial damage related to SARS-CoV-2 infection associated with increased mortality [15] with a particular increase in in-hospital mortality in patients with ST-segment elevation myocardial infarction (STEMI) [12,16]. The inflammatory pathophysiological mechanisms causing plaque rupture and the production of a prothrombotic environment may potentially increase the risk of local micro thromboembolism, may impair reperfusion, and increase the risk of coronary in-stent thrombosis [17]. This effect is due to a COVID-19 cytokine storm resulting from an imbalance in T-cell activation with dysregulated release of interleukin (IL)-6, IL-17, and other cytokines [14]. Because the inflammatory response is self-amplifying, the early diagnosis and attenuation can potentially have a significant impact on outcomes [18]. In-stent thrombosis could also potentially be a consequence of the larger thrombus burden and specific mechanisms associated with plaque disruption among SARS-CoV-2 positive patients, which could lead to an increased risk of thrombus formation and coronary thrombotic complications [14,19].

Moreover, several studies pointed out the crucial role of lipid rafts during viral infection [20,21,22]. Lipid rafts are microdomains within the plasma membrane that are rich in sphingolipids and cholesterol [23], play a key role in immunity and inflammation [24] and are involved in regulating the platelet function activation [21,25]. Lipid rafts are also the sites of the initial binding, activation, internalization, and cell-to-cell transmission of SARS-CoV-2 [21,22]. SARS-CoV-2 infection relies on the binding of S protein (Spike glycoprotein) to ACE (angiotensin-converting enzyme) 2 in the host cells [20]. Antiplatelets drugs are tested against the Spike proteins using in silico methods [26]. Thus, therapeutic approaches targeting lipid rafts may mitigate both COVID-19 infection itself and thromboinflammatory platelet response [20,21].

The published cases raise concerns about the increase in platelet aggregation as well as other aspects of prothrombotic state associated with COVID-19, leading to an increased risk of all types of stent thrombosis, from acute to very late. We present the biggest worldwide analysis and review of coronary stent thrombosis cases related to COVID-19.

## 2. Methods

We conducted a comprehensive literature search of studies using PubMed, Google Scholar, and Web of Science databases to identify the relevant literature. This review covers all published cases of COVID-19 stent thrombosis from the outbreak of the coronavirus pandemic to the end of year 2021. The keywords were: COVID-19; SARS-CoV-2; coronavirus; stent thrombosis; in-stent thrombosis; COVID-19 stent occlusion.

## 3. Results

All 17 patients were included in the analysis. Median age: 65 years (min.: 39; max.: 86; interquartile range: 14.0), 100% of patients were male. On admission, 14 (82.4%) patients presented STEMI, while 3 (17.6%) patients presented non-ST-segment elevation myocardial infarction (NSTEMI). The incidence of comorbidities was not different from other populations of patients with acute coronary syndrome diagnosis and was as follows: diabetes mellitus 41.2%, chronic kidney disease 11.8%, hypertension 52.9%. Patients presented all types of stent thrombosis: acute (4 patients; 23.5%), subacute (4 patients; 23.5%), late (1 patient; 5.9%) with a definite predominance of the very late-type (8 patients; 47.1%). In more than half of the patients, thrombosis occurred in the stent localized in the left anterior descending artery (LAD), in 5 (29.4%) patients in right coronary artery (RCA) and in 3 (17.6%) patients in the left circumflex artery. One report presented a rare case of a dual coronary stent thrombosis in LAD and RCA. Glycoprotein IIb/IIIa (GP IIb/IIIa) inhibitors were used in 8 (47.1%) patients and a balloon angioplasty was performed in 11 (64.7%) cases. 7 (41.2%) patients were treated with thrombectomy and a placement of new drug-eluting stent occurred in 6 (35.3%) patients. Endovascular imaging techniques were used in 3 (17.6%) cases. One procedure was supported with an Impella mechanical circulatory device. The mortality rate in this analysis was 35.3%. Patient characteristics and procedure specifications are presented in Table 2 and Table 3.

## 4. Discussion

Stent thrombosis is more frequent in patients with multiple comorbidities and in patients with complex atherosclerotic lesions, especially in those with acute coronary syndrome (ACS), diabetes, chronic kidney disease, diffuse and bifurcation lesions, and small arteries requiring more than one stent [4,41]. A greater predisposition to thrombosis, both arterial and venous during COVID-19 has been established [42]. During SARS-CoV-2 infection, cytokine storm occurs 5 to 10 days after the beginning of symptoms, resulting in endothelial damage, activation of the platelets, and coagulation cascade. The presence of a stent in the coronary artery should be considered as a local stasis factor, which would complete the Virchow triad.

Although the number of 17 cases may seem small, the rate at which stent thrombosis occurs must be taken into account. One of the largest studies involving 2797 patients randomized in clinical trials reported only 68 cases of stent thrombosis over 4 years of follow-up [43]. In this pre-COVID study the mortality rate in the stent thrombosis group was 30.9% [43], indicating the importance of the issue.

In cases presented with ACS due to plaque rupture, dual antiplatelet therapy and full-dose anticoagulation drugs should be administered, unless contraindications exist [44,45,46,47]. It is worth mentioning that the choice of an appropriate dual antiplatelet therapy is also under discussion in patients with COVID-19 stent thrombosis. Some reports support ticagrelor as a potent and reversible P2Y12 inhibitor, while others prefer the use of prasugrel owing to its high potency and lack of interaction with COVID-19 therapies [28,30,31,34,38]. Expert recommendations have been published on antithrombotic therapy management, with special consideration given to possible interactions with the drugs used for COVID-19 [48]. Due to the inhibitory effect of lopinavir/ritonavir on CYP3A4, many drugs, especially new oral anticoagulants and clopidogrel or ticagrelor, should be prescribed with great caution [49]. Therefore, prasugrel may be the medication of choice in patients without contraindications. In the collected cases, there was no homogeneous management of antiplatelet therapy during PCI, in cases 1, 2, 3, 8, 10, 12 patients received clopidogrel, in cases 6, 11, 15, 17 ticagrelor, and in 4, 5 prasugrel, however, there was a tendency towards switching to ASA + ticagrelor after hospital discharge. Particular attention should also be paid to drug-to-drug interactions between antiplatelets or anticoagulants and COVID-19 investigational therapies. Parenteral antithrombotic drugs generally have no known major interactions with COVID-19 investigational therapies [48]. Because of the limited evidence on the effectiveness of the antiviral medications against SARS-CoV-2, antiplatelet therapy should be given priority in COVID-19 patients with ACS. Due to the high volume of thrombosis and the risk of no-reflow after PCI, aspiration thrombectomy and the use of injectable antiplatelets such as eptifibatide and tirofiban during the procedure or 24–48 h after the PCI is recommended by most interventional cardiologists [50,51,52,53]. Endovascular imaging techniques OCT or IVUS may also be helpful in diagnosing underlying atherosclerotic plaques.

Rodriguez-Leor et al.’s study showed an increase in-hospital mortality (23.1 vs. 5.7%), stent thrombosis (3.3 vs. 0.8%) and cardiogenic shock development after PCI (9.9 vs. 3.8%) in patients with STEMI and COVID-19 in comparison with contemporaneous non-COVID-19 STEMI patients [10]. Mechanical thrombectomy (44 vs. 33.5%, *p* = 0.046) and GP IIb/IIIa inhibitor administration (20.9 vs. 11.2%, *p* = 0.007) were more frequent in COVID-19 patients [10]. The numbers obtained in this case review also reveal this higher trend—mechanical thrombectomy was used in 41.2% patients, and 47.1% patients received GPIIb/IIIa inhibitors.

In observations from a large pre-COVID stent thrombosis registry RESTART mortality in the early and late stent thrombosis groups was 22.4 and 23.5%, respectively [54]. In other pre-COVID studies, the mortality in the stent thrombosis group was up to 30.9% [43]. The mortality rate of 35.3% presented in this analysis is higher than the above results, however, we must consider that a COVID-19 infection also carries its own risk of mortality. In this review, three patients died from multi-organ failure and three died from cardiac reasons.

## 5. Limitations

This study is a review of stent thrombosis cases in patients with COVID-19 infection reported worldwide. Although we have been able to contact some of the authors, lack of certain information and small group size are limiting factors.

## 6. Conclusions

Stent thrombosis is a rare but usually catastrophic event, frequently associated with large MI or death. SARS-CoV-2 infection activates inflammatory mechanisms that potentially create a prothrombotic environment and increases the risk of local micro thromboembolism and all types of stent thrombosis. In patients after PCI, with active COVID-19 infection and symptoms of acute coronary syndrome, the possibility of stent thrombosis should be considered. Further investigations focusing on optimal antithrombotic therapy in coronary artery disease patients presenting with COVID-19 are required.

## Figures and Tables

**Table 1 viruses-14-00260-t001:** Categories of stent thrombosis.

Category	Early		Late	Very Late
	Acute	Subacute		
Time after stent implantation	<24 h	>24 h but ≤30 days	>30 days but ≤1 year	>1 year

**Table 2 viruses-14-00260-t002:** Patients characteristics and procedure specifications.

Case Number	1	2	3	4	5	6	7	8
Publication	Prieto-Lobato et al. [27]				Hinterseer et al. [28]	Lacour et al. [29]	Antuna et al. [30]	Ayan et al. [31]
Gender	M	M	M	M	M	M	M	M
Age	39	71	86	85	65	68	81	64
Presentation	STEMI	NSTEMI	STEMI	STEMI	STEMI	STEMI	STEMI	NSTEMI
Localisation of stent thrombosis	LCX	RCA med.	LAD prox.	LAD prox.	LAD	LAD prox.	LAD	LCX (OM)
Type of thrombosed stent	DES ×2 3.0 × 15 mm	BMS 3.5 × 18 mm	DES 3.0 × 8 mm	DES 3.5 × 32 mm	DES 3.0 × 22 mm	DES	DES 3.0 × 15 mm	DES 3.0 × 38 mm
Type and time of stent thrombosis	Acute—30 min	Very late— 13 years	Very late— 2 years	Very late— 4 years	Very late— 2 years	2×: Acute—2 h and 36 h—Two Episodes	Late— 3 months	Subacute— 3 days
APT before admission	None	ASA	ASA	ASA	ASA	-	ASA + Clopidogrel	-
APT during PCI	ASA + Clopidogrel	ASA + Clopidogrel	ASA + Clopidogrel	ASA + Prasugrel	ASA + Prasugrel	ASA + Ticagrelor	N/A	ASA + Clopidogrel
Fibrynolytic therapy	-	-	-	-	-	Tenecteplase—due to extended delay to primary PCI	-	-
PCI technique:	GPIIb/IIIa BA OCT	GPIIb/IIIa DES Thrombectomy	DES	GPIIb/IIIa BA IVUS Thrombectomy	GPIIb/IIIa BA DES	BA Thrombectomy	GPIIb/IIIa BA OCT Thrombectomy	GPIIb/IIIa BA DES
APT discharge	ASA + Ticagrelor	ASA + Ticagrelor	ASA + Clopidogrel	ASA + Clopidogrel	ASA + Prasugrel	N/A	ASA + Ticagrelor	ASA + Ticagrelor
EF%	45	55	45	30	35	15	N/A	45
Patients characteristics:								
	DM prior ACS HTN	CKD prior ACS HTN	DM CKD prior ACS PVD HTN	prior ACS	DM prior ACS HTN	DM	prior ACS HTN	HTN
So2 (%) on admission	90	96	95	95	78	N/A	N/A	83
Significant elevation:								
D-Dimer	H	H	H	H	-	-	H	H
Fibrinogen	H	N	H	N	-	-	H	-
APTT	N	N	N	N	-	-	-	N
CRP	H	H	H	H	H	H	H	H
Ferritin	H	H	N	H	-	-	-	H
Follow-up	Survived: Discharged after 4 days	Survived: Discharged without complications	Survived: Discharged after 5 days	Survived: Discharged after 10 days	Death: ARDS -Multi-organ failure due to COVID-19 complications	Death: Recurrent stent thrombosis 36 h later. Thrombectomy failed to provide reperfusion	Survived: Discharged after 2 days	Survived

M—male; STEMI—ST-elevation myocardial infarction; NSTEMI—non-ST-segment elevation myocardial infarction; RCA—right coronary artery; LAD—left anterior ascending; LCX—left circumflex artery; OM—obtuse marginal artery; DES—drug eluting stent; BMS—bare-metal stent; APT—antiplatelet therapy; PCI—percutaneous coronary intervention; N/A—not available; GPIIb/IIIa—glycoprotein IIb/IIIa; BA—balloon angioplasty; OCT—optical coherence tomography; IVUS—intravascular ultrasound; EF—ejection fraction; ACS—acute coronary syndrome history; DM—diabetes mellitus; HTN—hypertension; CKD—chronic kidney disease; PVD—peripheral vessel disease; ARDS—acute respiratory distress syndrome; H—high level (above normal range); N—in normal range.

**Table 3 viruses-14-00260-t003:** Patients characteristics and procedure specifications.

Case Number	9	10	11	12	13	14	15	16	17
Publication	Galleazzi et al. [32]	Choudhary et al. [33]	Kunal et al. [34]	Elkholy et al. [35]	Hauguel-Moreau et al. [36]	Eskandarian et al. [37]	Kumar et al. [38]	Zaher et al. [39]	Tabatabai et al. [40]
Gender	M	M	M	M	M	M	M	M	M
Age	79	64	40	48	65	65	64	51	57
Presentation	STEMI	NSTEMI	STEMI	STEMI	STEMI	STEMI	STEMI	STEMI	STEMI
Localisation of stent thrombosis	RCA prox.	RCA	LAD prox.	LAD med.	LAD med. + RCA (PDA) —DUAL thrombosis	LAD	RCA	LCX	LAD
Type of thrombosed stent	DES	DES	DES 3.0 × 26 mm	DES 2.75 × 34 mm	N/A	DES	DES	2× DES 2.5 × 18 mm & 3.0 × 23 mm	DES 3.0 × 18 mm
Type and time of stent thrombosis	Very late—2 years	Subacute— 5 days	Very late— 2 years	Subacute— 3 days	Very late— 2 and 10 years	Very late—2 years	Acute— 2 h	Acute—minutes	Subacute— 26 days
APT before admission	ASA	-	ASA	-	ASA	ASA	-	ASA + Ticagrelor	ASA + Clopidogrel
APT during PCI	N/A	ASA + Clopidogrel	ASA + Ticagrelor	ASA + Clopidogrel	ASA + Cangrelor	N/A	ASA + Ticagrelor	N/A	ASA + Ticagrelor
Fibrynolytic therapy	-	-	-	-	-	+	-	-	-
PCI technique:	DES	Thrombectomy	GPIIb/IIIa BA	BA IMPELLA	DES	-	BA Thrombectomy	BA	GPIIb/IIIa BA Thrombectomy
APT discharge	N/A	N/A	ASA + Ticagrelor	N/A	ASA + Clopidogrel	N/A	ASA + Prasugrel	N/A	ASA + Ticagrelor
EF%	N/A	N/A	35	40	25	25	30	40	30
Patient characteristics:									
	prior ACS	-	prior ACS	DM HTN	-	-	-	DM prior ACS HTN	DM HTN
So2 (%) on admission	N/A	N/A	95	96	N/A	82	N/A	91	97
Significant elevation:									
D-Dimer	-	-	H	H	H	-	-	-	-
Fibrinogen	-	-	-	-	H	-	-	-	-
APTT	-	-	-	N	-	-	-	-	-
CRP	-	-	H	H	H	H	H	-	H
Ferritin	-	-	-	H	-	-	-	-	H
Follow-up	Death: Acute respiratory failure	Death: due to time delay and COVID-19 complications	Survived: discharged in a stable condition	Death: ventricular fibrillation refractory to cardioversion and amiodarone	Survived	Survived	Survived: discharged 6 days later	Death: deterioration after acute restenosis with subsequent sudden cardiac arrest	Survived: discharged after completing the isolation period

STEMI—ST-elevation myocardial infarction; NSTEMI—non-ST-segment elevation myocardial infarction; RCA—right coronary artery; LAD—left anterior ascending; LCX—left circumflex artery; DES—drug eluting stent; N/A—not available; APT—antiplatelet therapy; PCI—percutaneous coronary intervention; BA—balloon angioplasty; GPIIb/IIIa—glycoprotein IIb/IIIa; EF—ejection fraction; ACS—acute coronary syndrome history; DM—diabetes mellitus; HTN—hypertension; H—high level (above normal range); N—in normal range.

## Data Availability

Not applicable.

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
