# Peer review of "Coronary Stent Thrombosis in COVID-19 Patients: A Systematic Review of Cases Reported Worldwide"

_viruses, 2022, doi:10.3390/v14020260_

Round 1
Reviewer 1 Report
I read with great interest the article by Skorupski WJ et al:" Coronary stent thrombosis in COVID -19 patients: a systematic review of reported cases reported worldwide ". The article is well written and provides a comprehensive overview of the incidence of in-stent thrombosis, mortality rates, and treatment problems in COVID -19 patients. In particular, it provides additional information on potential interactions between COVID-19 drugs and antiplatelet agents and anticoagulants. The article is of great interest and provides important insights for the academic community in the field of cardiology.
Author Response
We sincerely thank the Reviewer for valuable comments.
Reviewer 2 Report
This review entitled “Coronary stent thrombosis in COVID-19 patients: a systematic review of cases reported worldwide” the authors focused their attention on stent thrombosis complication related to COVID-19.
The present review article is well written, and is of potential interest to the readers of Viruses. I recommend the publication of this work only after the authors have addressed the following minor comment:
Since the lipids play an essential role in platelet functions the authors should consider adding in the introduction a brief description of the role of lipid raft components in the recognition and interaction between the spike protein of SARS-CoV-2 and the host’s ACE2 receptor to underline the anti-platelets therapy in stent thrombosis as suggested by several studies and recent review articles (doi.org/10.3390/ijms22063180; doi:10.3389/fcell.2020.618296).
Author Response
We sincerely thank the Reviewer for valuable comments.
Minor comment: Since the lipids play an essential role in platelet functions the authors should consider adding in the introduction a brief description of the role of lipid raft components in the recognition and interaction between the spike protein of SARS-CoV-2 and the host’s ACE2 receptor to underline the anti-platelets therapy in stent thrombosis as suggested by several studies and recent review articles (doi.org/10.3390/ijms22063180; doi:10.3389/fcell.2020.618296).
Response: We have added information about the role of lipid raft and interaction between the Spike protein of SARS-CoV-2 and the ACE-2 receptor to the manuscript:
“Moreover, several studies pointed out the crucial role of lipid rafts during viral infection [20][21][22]. Lipid rafts are microdomains within the plasma membrane that are rich in sphingolipids and cholesterol [23], play a key role in immunity and inflammation[24] and are involved in regulating the platelet function activation[21][25]. Lipid rafts are the sites of the initial binding, activation, internalization and cell-to-cell transmission of SARS-CoV-2. [21][22]. SARS-CoV-2 infection relies on the binding of S protein (Spike glycoprotein) to ACE (angiotensin-converting enzyme) 2 in the host cells [20]. Antiplatelets drugs are tested against the Spike proteins using in silico methods[26]. Thus, therapeutic approaches targeting lipid rafts may mitigate both COVID-19 infection itself and thromboinflammatory platelet response [20][21].”
Reviewer 3 Report
The manuscript of Skorupski and colleagues focuses on the effect of COVID19 on coronary stenting, which is an interesting and timely subject. The troubling aspects of this study are listed below:
- The main question that makes this manuscript interesting at a first glance remains unanswered: are COCIV19 patients more exposed to stent thrombosis? The description of 17 cases does not tell us the prevalence of this event in COVI19 patients compared to non-COVID19 population.
- The number of cases is extremely small (i.e. 17 cases described in 13 previous studies) and the robustness of the conclusions of this study has to be questioned.
- The authors make little effort to compare the data for the 17 COVID19 cases of stent thrombosis with those of non-COVID19 patients. Neither have they compared the data with pre-COVD19 studies effectively. The result is that this manuscript provides very little information.
- There is no take-home message from this study. Why do we need to read this manuscript?
- One of the studies cited (i.e. Rodriguez-Leor et al., 2021) provides the most interesting information for a reader. So, it seems superfluous to read the current study. It seems advisable to read the Rodriguez-Leor study instead.
Author Response
We sincerely thank the Reviewer for valuable comments.
Response: In-stent thrombosis occurrence increased during the COVID-19 pandemic, however, there is still lack of comprehensive study describing this population. This review and worldwide analysis of coronary stent thrombosis cases related to COVID-19 summarize all available data.
Rodriguez-Leor et al. 2021 study described 91 COVID-19 STEMI patients, in 3.3% of patients (3 patients) coronary stent thrombosis occurred. Although the number of 17 cases in this review may seem small, the rate at which stent thrombosis occurs must be taken into account. One of the largest studies involving 2,797 patients randomized in clinical trials reported only 68 cases of stent thrombosis over 4 years of follow-up [Mauri et al.] .
References:
- Mauri, L.; Hsieh, W.; Massaro, J.M.; Ho, K.K.L.; D’Agostino, R.; Cutlip, D.E. Stent Thrombosis in Randomized Clinical Trials of Drug-Eluting Stents. Engl. J. Med. 2007, 356, 1020–1029, doi:10.1056/NEJMoa067731.
Round 2
Reviewer 3 Report
Unfortunately, the manuscript has changed very little from the previous version. I still do not find it useful or enjoyable to read, but it may be that my background differs too much from the authors and my opinion may not be relevant in this case. Therefore, I leave any decision to the editor, but I have to confirm my initial decision of "Rejection".